# Area Wide Monitoring of Plant and Honey Bee (*Apis mellifera*) Viruses in Blueberry (*Vaccinium corymbosum*) Agroecosystems Facilitated by Honey Bee Pollination

**DOI:** 10.3390/v15051209

**Published:** 2023-05-20

**Authors:** Eunseo Lee, Raj Vansia, James Phelan, Andrea Lofano, Adam Smith, Aiming Wang, Guillaume J. Bilodeau, Stephen F. Pernal, M. Marta Guarna, Michael Rott, Jonathan S. Griffiths

**Affiliations:** 1Department of Biology, University of Waterloo, Waterloo, ON N2L 3G1, Canada; 2London Research and Development Centre, Agriculture and Agri-Food Canada, 4902 Victoria Ave N, Vineland Station, ON L0R 2E0, Canada; 3Department of Biological Sciences, Brock University, St. Catharines, ON L2S 3A1, Canada; 4Sidney Laboratory, Centre for Plant Health, Canadian Food Inspection Agency, 8801 East Saanich Rd., North Saanich, BC V8L 1H3, Canada; 5London Research and Development Centre, Agriculture and Agri-Food Canada, 1391 Sandford Street, London, ON N5V 4T3, Canada; 6Ottawa Plant Laboratory, Canadian Food Inspection Agency, 3851 Fallowfield Rd., Ottawa, ON K2J 4S1, Canada; 7Beaverlodge Research Farm, Agriculture and Agri-Food Canada, P.O. Box 29, Beaverlodge, AB T0H 0C0, Canada

**Keywords:** virus, honey bee, metagenomics

## Abstract

Healthy agroecosystems are dependent on a complex web of factors and inter-species interactions. Flowers are hubs for pathogen transmission, including the horizontal or vertical transmission of plant-viruses and the horizontal transmission of bee-viruses. Pollination by the European honey bee (*Apis mellifera*) is critical for industrial fruit production, but bees can also vector viruses and other pathogens between individuals. Here, we utilized commercial honey bee pollination services in blueberry (*Vaccinium corymbosum*) farms for a metagenomics-based bee and plant virus monitoring system. Following RNA sequencing, viruses were identified by mapping reads to a reference sequence database through the bioinformatics portal Virtool. In total, 29 unique plant viral species were found at two blueberry farms in British Columbia (BC). Nine viruses were identified at one site in Ontario (ON), five of which were not identified in BC. Ilarviruses blueberry shock virus (BlShV) and prune dwarf virus (PDV) were the most frequently detected viruses in BC but absent in ON, while nepoviruses tomato ringspot virus and tobacco ringspot virus were common in ON but absent in BC. BlShV coat protein (CP) nucleotide sequences were nearly identical in all samples, while PDV CP sequences were more diverse, suggesting multiple strains of PDV circulating at this site. Ten bee-infecting viruses were identified, with black queen cell virus frequently detected in ON and BC. Area-wide bee-mediated pathogen monitoring can provide new insights into the diversity of viruses present in, and the health of, bee-pollination ecosystems. This approach can be limited by a short sampling season, biased towards pollen-transmitted viruses, and the plant material collected by bees can be very diverse. This can obscure the origin of some viruses, but bee-mediated virus monitoring can be an effective preliminary monitoring approach.

## 1. Introduction

Viruses are widespread in plant agricultural systems, but approaches towards virus detection are often biased for symptomatic plants and limited to serological or PCR-based detection assays of individual viruses [1,2]. Randomized field sampling can be used for unbiased early detection of disease agents, but is rarely implemented or limited to only a few pathogens. Metagenomics-based approaches applied towards farm-wide or area-wide agricultural monitoring could offer advantages over traditional pathogen surveys through identification of a wider diversity of pathogens from multiple individuals simultaneously [2,3,4,5,6]. Understanding the complexity of pathogens present at an ecosystem level could help to identify priorities for management [7]. The use of high throughput sequencing (HTS) approaches applied towards bulked plant issues, or through sampling insects, is becoming more prevalent [7,8,9,10].

Many agricultural systems depend on bee pollination activities for food production, particularly long-lived perennial fruit production systems including tree fruit and small berry crops [11,12]. The honey bee, *Apis mellifera* L., is the most widely used and important insect pollinator for global crop production [13,14]. Honey bees can forage within a radius of 1.5 km from their hives in agricultural settings, but have been shown to travel as far as 12–14 km in certain landscapes [15,16,17]. Foraging ranges may be reduced to less than 1 km in areas of intense agricultural production, where flower resources are high [16,18,19]. During foraging trips, plant material, including pollen and nectar, is collected by bees and transported to a centralized hive location. Bee-collected plant samples combined with HTS approaches can allow for the detection of multiple pathogens, and potentially provide a representative view of pathogens present in the immediate area [7,20].

Blueberry (*Vaccinium sect. Cyanococcus*) is a major fruit crop in Canada pollenated by commercial honey bees. Highbush varieties (*Vaccinium corymbosum*) are cultivated primarily in Ontario (ON) and British Columbia (BC), and lowbush varieties (*Vaccinum angustifolium*) in Quebec and the Atlantic provinces [21]. Blueberries are commonly pollinated by a number of native pollinators, while commercial honey bees and occasionally commercially reared bumble bees (*Bombus* spp.) are used on blueberry farms to improve fruit set and production [12,22,23]. Honey bees are typically housed in wooden langstroth boxes containing frames of beeswax cells, and can house up to 40,000–60,000 bees [24,25]. Forager (worker) bees visit multiple flowers and individual plants during foraging, providing diverse sources of inputs [26]. Bee bread, a fermented food source created from nectar and pollen, is collected by foragers and combined with bee saliva and honey in individual brood cells [27,28,29]. Bee bread represents a preserved pollen derivative through the action of lactic acid bacteria in synergy with the antimicrobial properties of honey [28,29], providing a rich source of virus-contaminated plant material from a variety of sources.

Flowers are central hubs for pathogen transmission between plants, and between pollinator species [30,31,32,33]. Bees are susceptible to a wide range of viruses that have been implicated in colony health and survival [34,35,36,37,38,39]. Some bee-infecting viruses including deformed wing virus (DWV, genus *Iflavirus*) and black queen cell virus (BQCV, genus *Triatovirus*) can be deposited on flowers for horizontal transmission between forager bees [40]. Many plant viruses are pollen-transmitted, in particular viruses in the genus *Ilarvirus* and *Nepovirus* [30,41]. Some of these viruses can cause serious disease in blueberry production systems, including blueberry shock virus (BlShV, genus *Ilarvirus*), tomato ringspot virus (ToRSV; genus *Nepovirus*) and tobacco ringspot virus (TRSV, genus *Nepovirus*) [2,5,42,43,44,45]. Blueberry scorch virus (BlScV, genus *Carlavirus*) is a major issue in blueberry production systems, particularly the Pacific Northwest, but it is not known to be pollen transmitted [44]. These pathogens, and potentially many others, could be associated with pollen or bees and therefore be detectable through metagenomics-based approaches.

Genomic sequencing technologies have advanced dramatically in recent years, but are still not widely used for pest monitoring in agricultural systems due to higher costs [46,47]. However, methods that can combine fewer samples representative of local ecosystems and high throughput sequencing can be a powerful and cost effective tool for pest monitoring, and for understanding viral diversity. Here, we describe metagenomics-based area wide monitoring of plant- and bee-infecting viruses using plant material collected by bees from blueberry farms in ON and BC. Viral profiles were created for each site and sample type to better understand the ecology and distribution of viruses in Canadian blueberry production systems.

## 2. Methods

### 2.1. Sample Collection and RNA Extraction

Four different sample types were collected from managed bee colonies temporarily located in two independent blueberry (*Vaccinium corymbosum*) farms in BC (BC1 and BC2), and one in ON, in the summer of 2021, during blueberry bloom. Queen sources for the bee colonies were locally bred, and were mixtures of Italian and Carniolan-derived stocks. Blueberries were the primary crop grown at the ON site, but other small fruit crops were grown nearby including strawberry (*Fragaria x ananassa*) and raspberry (*Rubus idaeus*), while both BC sites were more monocultured. The BC sites were located in the Fraser valley, approximately 4.6 km apart, and both consisted of the variety “Hardy Blue” with BC1 having plants 30 years old, while BC2 had plants that were 20 years old. The four sample types consisted of the following: (1) 25 individual returning forager bees collected outside the hive and contained pollen in the corbicula; (2) approximately 10 mL of *pollen* collected using pollen traps (ApiHex Beekeeping supplies, Guelph, ON, Canada); (3) 25 individual *hive bees* (adult workers without visible signs of pollen on their bodies) collected from inside the hive; and, (4) approximately 20 mL of *bee bread* (stored pollen) collected from individual frames with a sterile spatula. Multiple hives were located at each farm site, and replicates were collected from separate colonies at each location. Four replicates of each sample type were collected in BC sites, while three replicates were collected in ON. In addition, two replicates of mixed flower and leaf samples (plant samples) were collected from 10–20 randomly selected blueberry plants at each site, located near the beehives. Sample names were created based on their location, plant species, and sample type and replicate (Ex. BCFV- BC Fraser Valley, site 1 or 2, or ON for Ontario, followed by crop type BB for blueberries, the sample replicate 1–4, followed by sample type). Forager bees were denoted by F, Pollen—P, Bee bread—B, and hive bee—H. Plant tissue samples were denoted by T). Total RNA (totRNA) was extracted from each hive-related sample using the spectrum total plant RNA extraction kit (Sigma Aldrich, ON, Canada), while dsRNA was extracted from composite plant samples following Kesanakurti et al. (2016) [48].

### 2.2. RNA Sequencing

Extracted totRNA was treated with an rRNA depletion step using the RiboMinus^TM^ Plant Kit for RNA-Seq (Invitrogen, Waltham, MA, USA) as per the manufacturer’s instructions. Ribo depleted totRNA and dsRNA HTS libraries were generated using the Illumina TruSeq Stranded mRNA Library Prep kit, following the manufacturer’s protocol, starting after mRNA selection steps [48]. An Illumina NextSeq500 was used to generate single-ended 75 base read files to each sample. Libraries were dual indexed using the IDT for Illumina TruSeq RNA UD Indexes (Illumina, San Diego, CA, USA) and normalized. Then, 24–32 sample libraries were pooled and sequenced using a NextSeq500 high output kit v2.5, 75 cycles (Illumina) which generated between 10 and 16 million reads, on average for each sample. RNAseq files were uploaded to the Sequence Read archive under the bioprojects number PRJNA967701.

### 2.3. Bioinformatics

HTS sample files were imported into Virtool [48] (www.virtool.ca, accessed on 30 September 2021) for sample management, quality control (QC) and data analysis. Reads passing QC using FASTQC (https://www.bioinformatics.babraham.ac.uk/projects/fastqc/, accessed on 30 September 2021), were mapped to known virus species in plant and bee virus databases updated December 2021 using the Pathoscope 2 pipeline [49]. Reads were aligned to representative isolates of all known plant viruses pulled from Genbank. Viruses with representative isolates receiving at least one mapped read then had reads mapped against all of their known isolates. Bowtie2 2.3.2 [50] was used in local mode for both rounds of read mapping with minimum score (–min-score) set to “L, 20, 1.0”, seed length (-L) to 15, and mismatches per seed (-N) set to 0. In the second round, the maximum number of alignments returned (-k) was set to 100. Reads matching viruses were also mapped to a host reference genome. Except for a 0 value for mismatches per seed (-N), default Bowtie2 parameters were used for mapping. Reads were eliminated from the analysis if they had a greater or equal alignment score to the host versus the virus. Multi-mapping reads were handled using a refactored derivative of the Pathoscope2 identification module, which exactly matches the output of the published module. Pathoscope2 makes the assumption that uniquely mapped reads indicate most likely true source genomes. Read values are fractionally reassigned from least likely source genomes to most likely [51]. Virus identification based on Virtool was used to create sample-specific pathogen profiles. For the purposes of this study, a minimum of 10% genome coverage was required for a virus species to be considered a positive detection from both totRNA and dsRNA samples. Sample profiles were combined to create site-specific profiles which included calculating the frequency of detection for each sample type at each site, and the average frequency of detection across all samples, average genome coverage, and viral reads per million (VRPM) for each virus detected across all samples from BC and ON. VRPM is similar to transcripts per million, and was calculated from the total number reads mapping to each individual virus, dividing by genome length of the virus in kilobasepairs (Kbps), and then normalized for the total number of reads in the sequencing run, per million (Appendix A). Due to inconsistencies of Apple hammerhead viroid detections, read counts and VRPM were manually annotated using Geneious prime version 11.0.14.1 (Biomatters Inc., Boston, MA, USA). Genbank accession numbers for reference sequences used for virus detection, metadata and calculations can be found in Appendix A.

### 2.4. Phylogenetics and Sequence Analysis

Using host genome-subtracted de novo assembled contigs for each sample, sequences were aligned to the BlShV, PDV, and BQCV reference sequence (Appendix A) using Geneious Prime. Samples with full coat protein sequence coverage were used for pairwise and phylogenetic analysis. Pairwise nucleotide distance comparisons were constructed using Geneious prime. Maximum likelihood phylogenetic trees were constructed using MEGA 11 with 1000 bootstrap replications [52]. Viral coat protein sequences identified in this study were uploaded to Genbank (Appendix A).

## 3. Results

### 3.1. Virus Detection from Managed Bee Hives in Canadian Blueberry Farms

Samples collected from managed bee hives from BC and ON blueberry farms were sequenced and viral profiles created for each site. RNAseq files were analyzed using Virtool to identify viruses infecting plants, and viral profiles including metadata for each sample was compiled (Appendix A). In BC, out of 32 collected samples, 29 plant viruses were identified (Table 1). The most frequently detected genus or virus family were seven virus species from the genus *Ilarvirus* followed by *Capillovirus* (*n* = 3), and *Betaflexivirus* family (*n* = 3). Initial virus identification was carried out at the species level, with no strain distinctions. BlShV was the most commonly detected virus in BC, present in 78% of samples examined, followed by prune dwarf virus (PDV, genus *Ilarvirus*; 72%), strawberry necrotic shock virus (SNSV, genus *Ilarvirus*; 50%), and cherry virus A (CVA, genus *Capilovirus*; 44%). BlShV was present in 100% of samples collected from the BC2 site, and approximately 50% of samples collected from BC1. Other small berry-associated viruses including blackberry chlorotic ringspot virus (BCRV, genus *Ilarvirus*; 22%), blueberry mosaic-associated virus (BlMaV, genus *Ophiovirus*; 16%), blueberry latent virus (BLV, genus *Amalgavirus*; 6%), and BlScV (genus *Carlavirus*; 6%) were also detected. Average VRPM were highest for BlScV and PDV, while average genome coverage was greatest for privet leaf blotch-associated virus (PLBaV, genus *Idaeovirus*, 78%), prunus virus F (PVF, genus *Fabavirus*; 78%), and CVA (63%) (Table 1).

Differences in viral profiles are observable between BC1 and BC2 (Table 1 and Figure 1c). On average, 3.8 and 5.9 viruses were detected per sample from BC1 and BC2, respectively. Some viruses were unique to one site. Lilac leaf chlorosis virus (genus *Ilarvirus*), cherry virus F (genus *Fabavirus*), BLV, PLBaV, camellia ringspot-associated virus 1 and 2 (genus *Prunevirus*), and camellia ringspot-associated virus 3 (genus *Capillovirus*), CVA, apple stem grooving virus (ASGV, genus *Capillovirus*), and citrus tatter leaf virus (genus *Capillovirus*) were only detected in BC2, while brassica rapa virus 1 (unclassified *Rhabdoviridae*), cycas necrotic stunt virus (genus *Nepovirus*), pyrus pyrifolia cryptic virus (genus *Deltapartitivirus*), grapevine-associated ilarvirus (GaIV; genus *Ilarvirus*), and actinidia virus X (genus *Potexvirus*) were only detected in BC1 samples (Figure 1a, Table 1).

Composite leaf and flower samples from 10 random blueberry plants near the bee hives were also collected, dsRNA extracted, and sequenced to identify viruses infecting blueberry (Table 2). Fewer viruses (*n* = 7) were detected in composite plant samples compared to bee-collected samples (*n* = 29; totRNA extracted), but five viruses were detected in both bee and plant samples including BLV, BlMaV, BlScV, BlShV, and brassica campestris chrysovirus 1 (BCCV1) (Figure 1a). Only two viruses, BLV and BlShV, were detected in BC1 plant tissues, while seven were detected in BC2 samples including BLV, BlShV, BlMoV, BlScV, helianthus annuus alphaendornavirus (genus *Endornavirus*), BrCaCV1, and bell pepper endornavirus (BPEV, genus *Endornavirus*) from BC2 (Figure 1a; Table 2). Average VRPM and genome coverage were generally higher in dsRNA extracted plant tissues compared to bee-collected samples (Table 1 and Table 2).

The plant virus profile from the ON site was quite different from BC, with fewer virus species detected. Nine viruses were identified in 12 samples. Two viruses in the genus *Nepovirus*, tomato ringspot virus (ToRSV) and tobacco ringspot virus (TRSV), were the most frequently detected viruses at 42 and 33%, respectively (Figure 1b; Table 3). Four viruses were detected in all three sites across both provinces: CVA (BC: 44%, ON: 17%), BCCV1 (BC: 34%, ON: 25%) prunus necrotic ringspot virus (PNRSV, genus *Ilarvirus;* BC: 25%, ON: 17%), and AHVd (BC: 22%, ON: 8%) (Table 1, Figure 1c). The plant tissue virome was also quite different from BC samples, and even bee-collected samples from ON (Figure 1b; Table 4); four viruses were detected in ON blueberry leaf and flower tissue including blueberry green mosaic-associated virus (BGMaV; genus *Vitivirus*), BlMaV, BLV, and blueberry virus A (BVA; unassigned *Closteroviridae* family), which were not observed in bee-collected samples (Figure 1b). BlMaV and BLV were not detected in ON bee samples, whereas in BC both viruses were detected in plant and bee samples.

Bee-infecting viruses were also identified from the same samples (Table 5). In BC samples, 10 viruses were identified with BQCV being detected in every sample tested. Varroa destructor virus 1 (VDV1; genus *Iflavirus*) and israeli acute paralysis virus (IAPV; genus *Aparavirus*) were common in BC samples (94% and 63%, respectively), but were absent in the ON samples tested. In ON, only four viruses were identified, with BQCV being detected in 92% of the samples tested (Table 6). Lake Sinai virus group (LSV; genus *Sinaivirus*), Sacbrood virus (genus *Iflavirus*), and apis mellifera filamentous virus (unclassified dsDNA virus) were detected in both BC and ON samples. LSV was the only virus detected in plant tissue samples from BC (Figure 2a). VDV1, IAPV, hobart bee virus 1 (unclassified *Picornavirales* order), apis rhabdovirus 1 (unclassified *Rhabdoviridae* family), deformed wing virus (genus *Iflavirus*) and varroa destructor virus 3 (genus *Iflavirus*) were all present in BC samples, but absent in ON (Figure 2c). Very few bee viruses were detected in plant leaf and flower tissue: Four BC and two ON plant samples tested positive for LSV, and BQCV was also detected in ON plant tissue samples (Table 5; Figure 2a; Appendix A).

### 3.2. Comparison of Viral Profiles from Bee Pollen, Forager Bees, Hive Bees and Bee Bread

Efficiency of virus detection was evaluated by calculating the average number of viruses identified per sample at all three sites. Each site was treated as an independent replicate, and samples were only compared within each site for statistical tests. Typically, plant virus diversity was significantly higher in bee bread samples relative to other sample types at both BC sites sample (Figure 3a; two-way ANOVA, *p* < 0.05). While the trend was consistent in samples collected at the ON site, bee bread plant virus species richness was not significantly different from that of pollen or hive bees at this site (Figure 3a). In BC1, bee bread tissues had an average of 6.5 plant viruses detected per sample, while other samples ranged from 2.75–3.25 (Figure 1a; ANOVA *p* < 0.05). In BC2, bee bread had an average of 9.25 viruses detected per sample, with other sample types ranging from 4.5–6.5 (Figure 1a). Bee bread consistently had the most diverse virome, and significantly more plant viruses were detected in this sample type, although this trend was not as pronounced in ON. When examining the diversity of bee-infecting viruses detected, forager bees and hive bees typically had significantly more viruses detected than other sample types. However, in BC2, a similar number of bee viruses were detected in the pollen sample (Figure 1b).

### 3.3. Viral Diversity in Bee-Collected Samples

A number of plant viruses were detected in relatively high frequency, and with good genome coverage and sequencing depth. Sequences corresponding to the coat protein regions of BlShV, PDV, and BQCV were further examined to better understand their diversity in bee-collected samples. BlShV is only known to infect *Vaccinium* hosts, while PDV is not known to infect *Vaccinium* spp. [43]. BlShV was prominent in both BC sites, detected in a total of 9 samples from site 1 and in all 16 samples from site 2, with an average of 57.5% genome coverage (Table 1). Ten samples in total had complete coverage of the BlShV coat protein region, and was used to create a consensus sequence for pair-wise sequence analysis and phylogenetic analysis (Figure 4A,B). Very little sequence variation was detected between different samples, with all BlShV CP sequences being 99.9–100% identical to each other in all bee collected samples, and ~98.5% identical to the reference sequence (Figure 4B). All BlShV CP sequences recovered from bee samples clustered together according to phylogenetic analysis, while the cranberry (*Vaccinium macrocarpon*) isolates formed a separate clade (Figure 4A).

PDV was identified in both BC sites with an average genome coverage of 52.3%, but was not detected in ON samples (Table 1). Complete PDV CP sequences were recovered from 10 different samples, and were more divergent than the BlShV CP sequences (Figure 5A,B). Complete coverage was mostly obtained from bee bread or hive bee samples, with only one forager bee sample having complete CP coverage (Appendix A). Following alignment of these sequences with other major PDV CP sequences from three separate phylogroups [53], pairwise distances were calculated and a phylogenetic tree was constructed. Eight samples clustered tightly together and were 97.4–99.2% identical to each other, and also clustered closely with a PDV isolate from a BC cherry tree that is located in phylogroup 2 (AF208741, 98.2–99.5% identity Figure 5A,B). Three other samples also branched within phylogroup 2, with 96–99.2% identity to the BC cherry isolate. BCFV2-BB-B2 sample clustered more closely with two PDV isolates from sweet cherry in the USA (AF208740, GU066792). A final isolate, BCFV1-BB-3B, clustered in phylogroup 1, and was 91.2–92.8% identical to other isolates from this site and 92.3% identical to the BC cherry isolate (Figure 5A,B) [54], suggesting multiple PDV isolates circulating in this system.

Of the known bee viruses, BQCV was detected in all bee-collected samples in BC, and in most samples in ON. This virus was also prominent in plant samples, suggesting a ubiquitous presence in these ecosystems (Table 5 and Table 6). Complete capsid protein 4 sequences were recovered from 14 samples, predominantly from forager and hive bee samples. BQCV capsid protein 4 sequences were generally highly similar to each other, ranging from 98.4–99.9% identity within bee-collected samples (Figure 6A). The lone ON isolate branched closely with BC isolates, suggesting low diversity of this virus across Canada (Figure 6B).

## 4. Discussion

The pollen virome is a largely untapped resource for virus identification and discovery. Recent reports using bee-mediated monitoring to survey for invasive virus species [7,9], and a pollen virome study of wild flowering plants, identified many similar viruses to this study including Ilarviruses ApMV, BlChRSV PNRSV, SNSV, TSV, and nepoviruses ToRSV and TRSV [41]. Bee viruses were also identified in the wild plant pollen virome including DWV [41]. The transmission of viruses through pollen in blueberries, and how this relates to crop and bee health has long been a major research interest [42,43,55,56,57,58]. Here, we report the first blueberry pollen and bee metavirome, with unique viral profiles based on sample type, individual farms, and geographic regions, demonstrating a powerful area-wide monitoring approach for agricultural pathogens.

### 4.1. Plant Virome

Unsurprisingly many pollen-associated small berry-infecting viruses were detected in this study. BlShV was the most prominent virus identified in BC, consistent with previous reports of this virus being prevalent in the Pacific Northwest and readily pollen transmitted [44]. Although not known to be pollen transmitted, BlScV is a serious pathogen in blueberry production systems and is primarily vectored by aphids [56]. BlScV was only detected twice in forager bee-collected samples, but was prominent in plant samples. Association of this virus with forager bees could be due to direct flower tissue contact as opposed to associations with pollen [56]. BlShV, BlScV, BlMaV and BLV were all detected in both bee and plant tissues, suggesting a good overlap between bee-related identification of plant viruses and viruses infecting the host of interest.

While there was good overlap between viruses detected in bee and plant samples at both BC sites, there was no such overlap between the viruses detected from samples collected from ON. ToRSV and TRSV were common in bee samples but absent in plants, while BLV, BVA, BlMaV, and BGMaV were identified in plant tissues, but not bee-collected samples. ToRSV and TRSV could have been missed in our random sampling of composite leaf and flower tissues, or they could have originated from other host plants [45,59]. ToRSV can infect a wide host range, and cause serious economic losses [60]. BGMaV is a member of the *Vitivirus* genus, while BVA is a member of the *Closterovirus* genus, and neither genus is commonly associated with pollen transmission [61,62]. BLV and BlMaV were both detected in bee-collected and plant samples in BC but were not detected in ON bee samples, only in plant samples. The number of reads mapping to BlMaV was much lower in ON plant tissues compared to BC, which could reflect differences in the levels of infection at each site. Further research is required to better understand the specificity and sensitivity of plant virus detection through bees relative to the proportion of plants infected by these viruses in the immediate area.

Ilarviruses, in particular, were very prominent and widely detected. Many identified ilarviruses are commonly associated with small berry production such as SNSV or BCRV, and are likely infecting hosts aside from blueberries, such as *Rubus* spp., in the immediate area. BCRV and GaIV have not previously been reported in Canada, further demonstrating the advantages of bee monitoring in detecting new and emerging pathogens [63,64,65]. However, BCRV and GalV were identified with low read counts and genome coverage, further research should confirm and validate their presence in these systems. Other identified ilarviruses are more commonly associated with tree fruits such as PNRSV and PDV, and have not been reported to directly infect *Vaccinium* plants, to the best of our knowledge. It is unlikely these sequences were derived from blueberry plants, and are more likely to be present in other plant species nearby. Curiously, there was very little sequence variation for recovered BlShV CP sequences, which could reflect a founder effect due to clonal propagation practices in blueberry production. Other ilarviruses have been reported to have low CP sequence diversity including American plum line pattern virus [53,66]. Very few BlShV sequences are available in GenBank, and little is known regarding the population diversity of BlShV. Other ilarviruses such as PDV did show more diverse sequences. RNA3 of PDV separates into three phylogroups [53], we identified PDV CP sequences corresponding to all three phylogroups, suggesting diverse origins of PDV and potentially multiple strains of this virus circulating in the environment.

We expected the greatest diversity of plant viruses to be detected in plant tissues (pollen, bee bread, and plant tissue), and more bee viruses to be identified in bee samples (forager bee, hive bee), and for the most part this was consistent. Bee bread displayed the highest diversity of plant viruses, and also the highest sequence diversity of individual viruses. This is likely due to bee bread being collected over the seasonal flowering period of different plant species. Many of the viruses identified in bee bread are often associated with apple or tree fruit production systems such as ASGV, citrus concave gum-associated virus (genus *Coguvirus*), cherry virus F (genus *Fabavirus*) and citrus virus A (genus *Coguvirus*) that may have originated from plants flowering concurrently, before or after blueberries. Bee bread represents a highly diverse collection of plant pollen samples, and could be useful for further study of plant virus diversity. Surprisingly, hive bees also showed an unexpected richness in plant virus species compared to the other sample types which in theory do not directly interact with plants, suggesting the potential for dissemination of plant viruses within a bee colony. BlScV and BlMaV were only identified in forager or hive bee samples, and could have been acquired through direct flower contacts, as opposed to pollen transmission.

### 4.2. Bee Virome

Bee health and monitoring the viral load of bee colonies are other important aspects of ecosystem health monitoring [9,67,68]. BQCV and LSV were common in BC and ON sites, while VDV1 and IAPV were frequently detected at both BC sites and absent in ON. BQCV and IAPV have previously been detected in pollen and pollen pellets, and can be foodborne transmitted [69,70]. LSV, DWV, and SBV have been found in pollen and pollen pellets [71,72,73]. Peculiarly, LSV was widely prevalent in bee samples but absent in bee bread, and yet was also identified in plant leaf and flower tissue. Fecal-oral routes have been suggested to play an important role in LSV dissemination, and its relation to plant food source is important [74]. One dsDNA virus, apis mellifera filamentous virus, was detected in both BC and ON. Surprisingly, multiple samples had over 60% genome coverage for this virus, demonstrating the ability to identify DNA viruses from totRNA extracted samples.

### 4.3. Metagenomics-based Monitoring of Plant Viruses

A large number of plant and bee viruses were detected widely in different sample types, sites, and regions. An arbitrary genome coverage threshold of 10% was established to remove low-confidence identifications. Other genomics-based studies have argued for genome coverage cut off levels between 10 and 15% for genomics, and 20% for metagenomics-based surveys [2,41,51]. Genome coverage threshold cut offs can be arbitrary; yet, with metagenomics studies, it is important to cast a wide net to understand the diversity of viruses within the local population, and could help with the identification of emerging viruses. The trade-off to using a lower threshold is a greater chance of false positives inherent to the method [75]. Other studies have employed similar biovigilance approaches through RT-PCR-based assays to detect major viruses of concern in *Prunus* production systems [76]. Metagenomics-based approaches can complement more focused assays through identification of all pollen-associated viruses, and provide valuable viral sequence diversity information.

As evidenced from the wide diversity of plant viruses identified, pathogen monitoring through bee-mediated pollination activities is an excellent approach to understand the diversity of viruses present in an ecosystem. Previous studies of blueberry viruses have identified ~12 high priority viruses in the US and BC [44]. Many of the viruses listed in this report were identified through bee/pollen screening including BlShV, BlScV, BLV, BlMaV, TRSV and ToRSV. Other viruses identified in plant tissue but not bee samples include BGMaV and BVA, which have not been previously reported in Canada. While many viruses are detectable in this system, the exact risks of transmission for each specific virus can be unclear. Understanding pollen transmission risks for each virus must still be individually determined. Bee-mediated virus monitoring can perform as efficiently or better than randomly collected samples while also providing useful data regarding sequence diversity, demonstrating the utility of a bee-mediated metagenomics-based pathogen monitoring approach in agricultural systems. Despite these advantages, it is important to understand the limitations of bee-mediated pathogen monitoring: only pollen-transmitted viruses or those contaminating bees acquired through interactions with flowers can be detected.

## Figures and Tables

**Figure 1 viruses-15-01209-f001:**
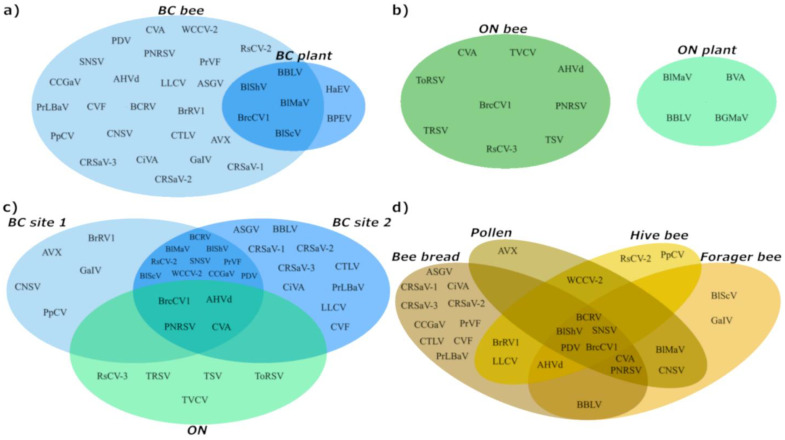
Venn diagram comparison of overlapping plant virome profiles from BC and ON blueberry farms. (**a**) Plant viruses detected in BC bee-related samples and plant-specific tissues. (**b**) Plant viruses detected in ON bee-related samples and plant-specific tissues. (**c**) Plant viruses detected in BC site 1, site 2, and the ON blueberry site from bee-related samples. (**d**) Plant viruses detected in bee bread, pollen, forager bee, and hive bee from both BC blueberry sites.

**Figure 2 viruses-15-01209-f002:**
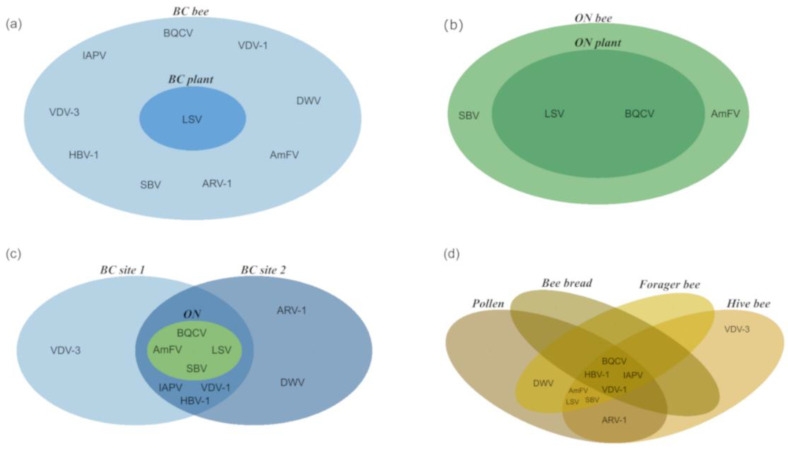
Venn diagram comparison of overlapping bee virome profiles from BC and ON blueberry farms. (**a**) Bee viruses detected in BC bee-related samples and plant-specific tissues. (**b**) Bee viruses detected in ON bee-related samples and plant-specific tissues. (**c**) Bee viruses detected from BC site 1, site 2, and the ON site in bee-related samples. (**d**) Bee viruses detected in bee bread, pollen, forager bee, and hive bee samples from both BC blueberry sites.

**Figure 3 viruses-15-01209-f003:**
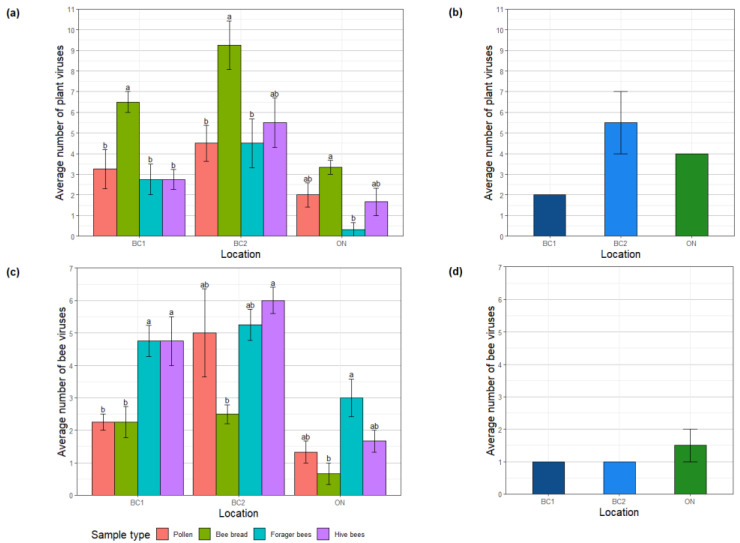
Virus species richness in different bee-related samples. (**a**) Average number of plant viruses identified in pollen, forager, bee bread, hive bee, and plant tissues from BC (*n* = 4) and ON (*n* = 3). (**b**) Average number of plant viruses identified in plant tissue from BC sites 1 and 2, and the ON site. (**c**) Average number of bee viruses identified in bee samples from BC (*n* = 4) and ON (*n* = 3) sites. (**d**) Average number of bee viruses identified in plant tissue from BC sites 1 and 2, and the ON site. Error bars indicate standard error. Letters correspond to categories of significant differences between samples based on a two-way ANOVA with Tukey’s post hock test (*p* < 0.05). Each site was analyzed independently, and statistical analysis was performed within the dataset for each site independently.

**Figure 4 viruses-15-01209-f004:**
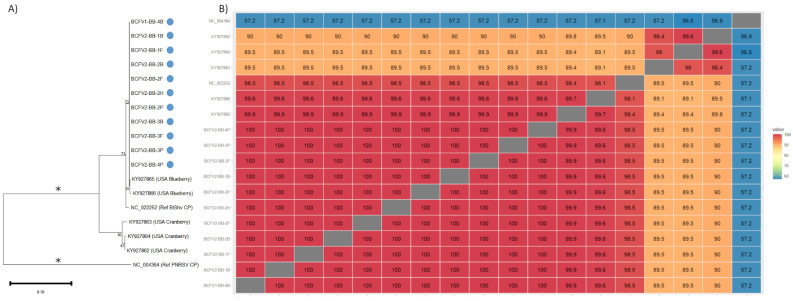
Pairwise nucleotide sequence identity and phylogenetic analysis of the coat protein region of BIShV from bee-collected samples in BC blueberry farms. (**A**) Maximum likelihood phylogenetic tree of BIShV CP nucleotide sequences. PNRSV CP was used as a outgroup. Asterisk indicates bar lengths are not to scale. (**B**) Pairwise nucleotide sequence identity comparison for BIShV CP nucleotide sequences, compared with sequences available on NCBI.

**Figure 5 viruses-15-01209-f005:**
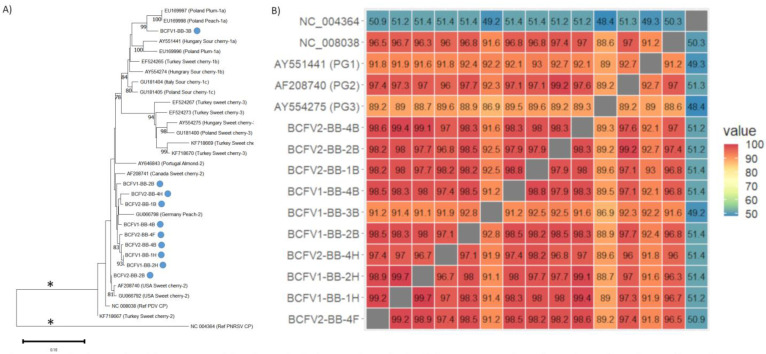
Pairwise nucleotide sequence identity and phylogenetic analysis of the coat protein region of PDV from bee-collected samples in BC blueberry farms. (**A**) Maximum likelihood phylogenetic tree of PDV CP nucleotide sequences. PNRSV CP was used as a outgroup. Asterisk indicates bar lengths are not to scale. (**B**) Pairwise nucleotide sequence identity for PDV CP nucleotide sequences, compared with the PDV CP refseq (NC_008038), representatives from Phylogroup 1 (AY551441), Phylogroup 2 (AF208740), Phylogroup 3 (AY554275), and PNRSV (NC_004364) was used as an outgroup.

**Figure 6 viruses-15-01209-f006:**
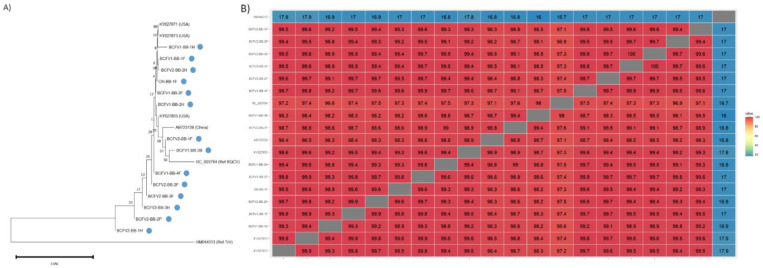
Pairwise nucleotide sequence identity and phylogenetic analysis of the capsid protein region of BQCV from bee-collected samples in Canadian blueberry farms. (**A**) Maximum likelihood phylogenetic tree of BQCV capsid nucleotide sequences. Triatoma virus capsid protein 4 sequence (HMO44313) was used as a outgroup. (**B**) Pairwise nucleotide sequence identity for BQcV capsid nucleotide sequences, compared with published sequences.

**Table 1 viruses-15-01209-t001:** Plant viruses detected in bee-collected samples from 2 BC blueberry farms.

Plant Virus	Genus	BC Site 1				BC Site 2				Average Frequency of Detection (%) *	Average Genome Coverage (%) *	Average VRPM *
		Bee Bread	Forager Bee	Hive Bee	Pollen	Bee Bread	Forager Bee	Hive Bee	Pollen			
		*n* = 4	*n* = 4	*n* = 4	*n* = 4	*n* = 4	*n* = 4	*n* = 4	*n* = 4			
		Frequency (%)										
Blueberry shock virus	Ilarvirus	75	50	50	50	100	100	100	100	78	57.5	425
Prune dwarf virus	Ilarvirus	100	50	75	50	100	50	100	50	72	52.3	356
Strawberry necrotic shock virus	Ilarvirus	75	25	0	75	50	50	50	75	50	56.6	281
Cherry virus A	Capillovirus	75	25	25	0	100	50	75	0	44	63.2	176
Brassica campestris chrysovirus 1	Alphachrysovirus	50	0	0	0	50	25	75	75	34	52.8	24
White clover cryptic virus 2	Betapartitivirus	0	0	50	50	0	0	75	25	25	24.5	43
Prunus necrotic ringspot virus	Ilarvirus	50	25	25	0	100	0	0	0	25	62.9	290
Blackberry chlorotic ringspot virus	Ilarvirus	25	0	0	0	25	25	50	50	22	17.4	0
Apple hammerhead viroid	Pelamoviroid	75	0	0	25	25	25	0	25	22	58.6	5
Blueberry mosaic-associated virus	Ophiovirus	0	25	0	0	0	75	25	0	16	34.3	40
Citrus concave gum-associated virus	Coguvirus	25	0	0	0	50	0	0	0	9	29.6	143
Prunus virus F	Fabavirus	50	0	0	0	25	0	0	0	9	77.7	16
Lilac leaf chlorosis virus	Ilarvirus	0	0	0	0	50	0	0	25	9	35.0	230
Brassica rapa virus 1	unclassified Rhabdoviridae	50	0	0	25	0	0	0	0	9	17.2	93
Blueberry latent virus	Amalgavirus	0	0	0	0	25	25	0	0	6	25.5	3
Blueberry scorch virus	Carlavirus	0	25	0	0	0	25	0	0	6	42.3	147
Cherry virus F	Fabavirus	0	0	0	0	50	0	0	0	6	27.1	0
Cycas necrotic stunt virus	Nepovirus	0	25	25	0	0	0	0	0	6	34.3	23
Raphanus sativus cryptic virus 2	unclassified Partitiviridae	0	0	0	25	0	0	0	25	6	31.3	44
Apple stem grooving virus	Capillovirus	0	0	0	0	25	0	0	0	3	24.0	0
Citrus tatter leaf virus	Capillovirus	0	0	0	0	25	0	0	0	3	12.9	0
Citrus virus A	Coguvirus	0	0	0	0	25	0	0	0	3	28.7	1
Pyrus pyrifolia cryptic virus	Deltapartitivirus	0	0	0	25	0	0	0	0	3	12.4	0
Privet leaf blotch-associated virus	Idaeovirus	0	0	0	0	25	0	0	0	3	77.9	44
Grapevine-associated ilarvirus	Ilarvirus	0	25	0	0	0	0	0	0	3	10.1	36
Actinidia virus X	Potexvirus	0	0	25	0	0	0	0	0	3	10.0	57
Camellia ringspot-associated virus 1	unclassified Betaflexiviridae	0	0	0	0	25	0	0	0	3	40.2	0
Camellia ringspot-associated virus 2	unclassified Betaflexiviridae	0	0	0	0	25	0	0	0	3	34.3	0
Camellia ringspot-associated virus 3	unclassified Betaflexiviridae	0	0	0	0	25	0	0	0	3	58.8	2

* Average frequency of detection, genome coverage and viral reads per million (VRPM) were calculated across all positive detections for all samples.

**Table 2 viruses-15-01209-t002:** Plant viruses detected in leaf and flower samples from 2 BC blueberry farms.

Plant Virus	Genus	BC Site 1	BC Site 2	Average Frequency of Detection (%) *	Average Genome Coverage (%) *	Average VRPM*
		Plant Tissue	Plant Tissue			
		*n* = 2	*n* = 2			
		Frequency (%)				
Blueberry latent virus	Amalgavirus	100	100	100	99.9	55,109
Blueberry shock virus	Ilarvirus	100	100	100	82.7	20,711
Blueberry mosaic-associated virus	Ophiovirus	0	100	50	99.6	625
Blueberry scorch virus	Carlavirus	0	100	50	66.3	900
Bell pepper endornavirus	Alphaendornavirus	0	50	25	25.2	1
Brassica campestris chrysovirus 1	Alphachrysovirus	0	50	25	17.2	0
Helianthus annuus alphaendornavirus	Alphaendornavirus	0	50	25	100.0	338

* Average frequency of detection, genome coverage and viral reads per million (VRPM) were calculated across all positive detections for all samples.

**Table 3 viruses-15-01209-t003:** Plant viruses detected in bee-collected samples from one ON blueberry farm.

Plant Virus	Genus	ON Site 1				Avearge Frequency of Detection (%) *	Average Genome Coverage (%) *	Average VRPM *
		Bee Bread	Forager Bee	Hive Bee	Pollen			
		*n* = 3	*n* = 3	*n* = 3	*n* = 3			
		Frequency (%)						
Tomato ringspot virus	Nepovirus	100	0	33	33	42	33.3	75
Tobacco ringspot virus	Nepovirus	67	0	67	0	33	22.9	61
Raphanus sativus cryptic virus 3	unclassified Partitiviridae	67	0	0	33	25	40.2	22
Brassica campestris chrysovirus 1	Alphachrysovirus	0	0	0	100	25	22.8	57
Cherry virus A	Capillovirus	33	0	0	33	17	16.8	60
Prunus necrotic ringspot virus	Ilarvirus	33	33	0	0	17	13.8	41
Apple hammerhead viroid-like circular RNA	Pelamoviroid	33	0	0	0	8	17.5	0
Turnip vein-clearing virus	Tobamovirus	0	0	33	0	8	11.0	67
Tobacco streak virus	Ilarvirus	0	0	33	0	8	10.7	128

* Average frequency of detection, genome coverage and viral reads per million (VRPM) were calculated across all positive detections for all samples.

**Table 4 viruses-15-01209-t004:** Plant viruses detected in leaf and flower samples from one ON blueberry farms.

Plant Virus	Genus	ON Site 1	Average Frequency of Detection (%) *	Average Genome Coverage (%) *	Average VRPM *
		Plant Tissue			
		*n* = 2			
		Frequency (%)			
Blueberry green mosaic-associated virus	Vitivirus	100	100	98.9	34
Blueberry latent virus	Amalgavirus	100	100	99.7	61,635
Blueberry mosaic-associated virus	Ophiovirus	100	100	68.8	7
Blueberry virus A	Unassigned (Closteroviridae family)	100	100	61.9	922

* Average frequency of detection, genome coverage and viral reads per million (VRPM) were calculated across all positive detections for all samples.

**Table 5 viruses-15-01209-t005:** Bee viruses detected in bee-collected samples from two BC blueberry farms.

Bee Virus	Genus	Bee Bread	Forager Bee	Hive Bee	Pollen	Bee Bread	Forager Bee	Hive Bee	Pollen	Average Frequency of Detection (%) *	Average Genome Coverage (%) *	Average VRPM *
		BC Site 1				BC Site 2						
		*n* = 4	*n* = 4	*n* = 4	*n* = 4	*n* = 4	*n* = 4	*n* = 4	*n* = 4			
		Frequency (%)										
Black Queen Cell Virus	Triatovirus	100	100	100	100	100	100	100	100	100	81.1	1301
Varroa destructor virus 1	Iflavirus	75	100	100	75	100	100	100	100	94	59.5	484
Israel Acute Paralysis Virus	Aparavirus	25	25	75	25	50	100	100	100	63	68.1	911
Lake Sinai Virus	Sinaivirus	0	100	100	0	0	100	100	75	59	88.6	5564
Apis mellifera filamentous virus	Unclassified dsDNA virus	0	75	25	0	0	50	75	25	31	46.8	16
Hobart bee virus 1	Unclassified Picornavirales	25	50	50	0	0	0	50	25	25	26.1	0
Sacbrood Virus	Iflavirus	0	25	0	25	0	50	50	25	22	51.6	762
Apis rhabdovirus 1	Unclassified Rhabdoviridae	0	0	0	0	0	0	25	25	6	70.1	0
Deformed Wing Virus	Iflavirus	0	0	0	0	0	25	0	25	6	65.3	4316
Varroa destructor virus 3	Iflavirus	0	0	25	0	0	0	0	0	3	12.7	0

* Average frequency of detection, genome coverage and viral reads per million (VRPM) were calculated across all positive detections for all samples.

**Table 6 viruses-15-01209-t006:** Bee viruses detected in bee-collected samples from one ON blueberry farm.

Bee Virus	Genus	ON Site 1				Average Frequency of Detection (%) *	Average Genome Coverage (%) *	Average VRPM *
		Bee Bread	Forager Bee	HIVE BEE	Pollen			
		*n* = 3	*n* = 3	*n* = 3	*n* = 3			
		Frequency (%)						
Black Queen Cell Virus	Triatovirus	67	100	100	100	92	64.7	140
Lake Sinai Virus	Sinaivirus	0	100	33	0	42	58.2	626
Sacbrood Bee Virus	Iflavirus	0	67	33	0	25	42.6	158
Apis mellifera filamentous virus	Unclassified dsDNA virus	0	33	0	0	8	79.6	31

* Average frequency of detection, genome coverage and viral reads per million (VRPM) were calculated across all positive detections for all samples.

## Data Availability

The data presented in this study are available on request from the corresponding author.

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
