# Peer review of "Area Wide Monitoring of Plant and Honey Bee (Apis mellifera) Viruses in Blueberry (Vaccinium corymbosum) Agroecosystems Facilitated by Honey Bee Pollination"

_viruses, 2023, doi:10.3390/v15051209_

Round 1

Reviewer 1 Report

The manuscript is well-written and describes some interesting work with plant viruses.  The study involved using a metagenomic approach on bee and plant samples from three blueberry farms in two Canadian provinces.  The authors used the PathoScope pipeline to screen for virus sequences within pollen, bee bread, plant and bee samples.  Results present sequence data of a wide variety of plant viruses presumably picked-up by bees as they forage on blueberries and other flowering plants.  Some common and expected sequences of known pollen-borne viruses in Ilarvirus and Nepovirus genera were identified as well as some rarer, unexpected, and largely unexplained sequences like the Blueberry scorch virus.  The biggest weakness of the study is a lack of any validation of the metagenomic results.  Some of the text suggests larger implications than the data support.  For example in Line 54 the statement suggesting a representative view of pathogens in the immediate area can be obtained from this type of study is misleading.  The authors later acknowledge some of the limitations (ex. The closing line of the manuscript), but I think some additional emphasis on the limitations of representation of bee/pollen samples, the timing and limited nature of sampling, and the relatively few replications and sites is needed.  That said, resources are limited and the work provided is sufficient for publication.  The corrections needed are minor, and this represents a nice contribution that I was pleased to have been asked to review.  Specific suggestions are included below.

Abstract:

“…associated with two blueberry farms in BC…”

Line 42 “pre-emptive biovigilance”?

Line 43 issues to tissues

Line 54 “…providing a representative view of pathogens present in the immediate area…”

Line 82 …”could contaminate pollen or bees and therefore be detectable through metagenomics-based approaches.”

Line 84 is to are

Line 95 Were the hives resident at the site permanently? Can you give additional details about the bees? Stock, race, etc.?

Line 109 Were leaf and flower samples washed or surface-sterilized prior to extraction? What was the age of the plants sampled? What varieties were sampled?

Line 128 the Patho scope 2 pipeline49

Line 129 reads is duplicated

Line 140 will be addressed in the discussion ? “Due to inconsistencies of Apple hammerhead viroid de- 140

tections, read counts and VRPM were manually annotated using Geneious prime version 141

11.0.14.1 (Biomatters inc, CA, USA).”

Line 153-7 duplication of M&M, could be eliminated

Line 185 was to were 

Line 217 …” the same RNAseq data files…” same as what? Which files are being compared here?

Fig5a Suggest highlighting, in some way, sequences from this study to make distinct from ref seqs used.

Line 363 “…represents a first report for these viruses in Canada…” First reports generally require two independent methods of confirmation.  Your study suggests these are present and offers one line of convincing evidence, but some level of validation is needed to make this statement definitively.  The same general comment applies throughout the discussion.  I don’t believe it’s necessary for publication of this study, but there should be some additional acknowledgement of the limitations of relying strictly on interpretation of metagenomic data.

Line 366 “have not been reported to directly infect Vac-cinium plants…” They are not pathogens of Vaccinium, and your data do not provide evidence to the contrary. 

Line 386-7 They are living in the hive where the pollen, nectar, and bee bread is made, stored, used. Were the hive bees surface sterilized with bleach prior to RNA extraction? External contamination?

Reviewer 2 Report

Title

The title is confusing

I am unsure what ‘Area wide monitoring’ is. Do you mean detection of a wide area?

I am also unsure what is ‘facilitated by honey bee pollination’.

The title should be a one-line description of what you found.   

Abstract.

Define ON and BC at first mention. This is an international journal.

I was puzzled why details of BIShV and PDV CP sequences were given, but none for other viruses.

The title suggests both plant and insect viruses were studied, but the bee viruses found, if any, are not referred to in the abstract.

The method used to find the viruses is not given in the abstract.

Introduction

The Intro section is overly wordy and repetitive in places. For example, ‘(line 51) During foraging trips, plant material including pollen and nectar are collected by bees and transported to a centralized hive location’ and ‘ (line 63) Forager (worker) bees visit multiple flowers and individual plants during foraging trips where they collect nectar and pollen to be transported back to the hive’  is more concisely written ‘Bees collect pollen and nectar which is stored in hives.’

Clarify ‘Bee-collected plant samples’

Is all the information about ‘bee bread’ (lines 65-70) necessary?

Line 102 - 105. How is it possible to collect 10 mL and 20 mL of pollen? Perhaps convert this to a weight rather than a volume? (Please place a space between the number and the unit, e.g. 10mL vs 10 mL)

No information is given on the preparation of the samples, how the libraries were made, the results of the sequencing (numbers, quality scores), and what was done to the raw sequences to remove primer-adaptor sequences. Were samples pooled or was each sample sequenced independently? If pooled, how were they separated into individual samples?

Software developers must be named, e.g. Virtool.

Presumably most viruses found were ssRNA genomes. Why were viral genomes measured in Kbps?

Table 1. Please provide information on the length of the genome detected. What does average genome coverage mean?

Please provide the accession numbers of all the sequences you discovered. The section ‘Accession number’ in Supp tables 1 and 2 are NOT accessions of your viruses. Where are they?

Supplementary files AVM S files 1 and 2

There are no legends. Present the Supp files in the format requested by the journal. Abbreviations are not defined. What do the numbers represent? For example, Blackberry chlorotic ringspot virus has 0 in several samples, but no hits. Why is it included?

Some virus x samples have large numbers (of reads matching this virus?) numbering in the tems of thousands, while others are in single digits. What does this mean? Is this discussed in the text?

What are the colours in the ‘Accession number ‘ pages?

Although it is clear the authors have done a lot of work, it is all wasted if the results are not more clearly shown, and all the virus sequences from each of the sites are available on public repositories.

I am recommending major corrections not because the data is insufficient, but because it is poorly described.

Round 2

Reviewer 2 Report

Thanks for your amendments. I am happy to recommend publication of this second iteration.